# Magneto-Viscoelastic Materials: Memory Functionals and Rate Equations

**DOI:** 10.3390/ma15196699

**Published:** 2022-09-27

**Authors:** Claudio Giorgi, Angelo Morro

**Affiliations:** 1Dipartimento di Ingegneria Civile, Architettura, Territorio, Ambiente e di Matematica, Università di Brescia, Via Valotti 9, 25133 Brescia, Italy; 2Dipartimento di Informatica, Bioingegneria, Robotica e Ingegneria dei Sistemi, Università di Genova, Via All’Opera Pia 13, 16145 Genova, Italy

**Keywords:** magneto-viscoelastic materials, magneto-viscoelasticity of strain rate type, memory functionals, rate equations, thermodynamic consistency

## Abstract

The properties of viscoelastic solids subject to a magnetic field are modelled within two thermodynamically consistent approaches that are typical of models with a non-instantaneous response. One is based on memory functionals: the reversible changes are described by the instantaneous response, while the dissipativity is expressed by the dependence on histories. The other approach involves objective rate equations. While memory functionals lead to the difficulty of determining thermodynamically consistent free energy functionals, rate equations result in a simpler scheme. The greater simplicity allows the discovery of, in particular, models of magneto-hyperelastic materials, magneto-hypoelastic materials, and various forms of magneto-viscoelastic behaviour. The novelty of the procedure is based on two features: a representation formula, originating from the entropy inequality, and the use of the entropy production as a constitutive function. Relations with other approaches in the literature are examined in detail.

## 1. Introduction

The response of magnetic materials to an applied magnetic field is generally not instantaneous. In addition to showing a partial instantaneous response, the material gradually approaches equilibrium in a finite time dependent on the deformation. Magneto-viscoelastic materials are intended to be models that exhibit both instantaneous changes of magneto-mechanical properties and a variable time response when acted upon by a magnetic field. This subject is of interest for applications and requires both appropriate balance equations and constitutive equations. Balance equations are treated, e.g., in [1,2], while interesting constitutive equations are developed in, e.g., [3,4]. Updated lists of references are given in [5,6,7]. Despite the various approaches and procedures developed in the literature, the subject deserves further attention, hopefully to create simpler models.

Recently, we have developed a systematic approach to nonlinear materials with memory [8,9]. The crucial points are that the models are thermodynamically consistent; the entropy production is a constitutive quantity that characterizes the dissipative properties; the entropy inequality allows us eventually to derive a representation formula for the pertinent function; and the nonlinear rate equations are derived with an objective time derivative.

In essence, magnetic viscoelasticity involves the interaction between mechanical and magnetic fields within dissipative processes in materials with a non-instantaneous response. This scheme is realized by letting the independent variables occur through their histories or by considering rate-type constitutive equations. Further, in both cases, the thermal properties are modelled, and the restrictions placed by objectivity and thermodynamic consistency are investigated in detail in order to be able to make a comparison between the characteristics of the two approaches.

While balance equations are used as the standard in the literature (see, e.g., [6,10,11]), constitutive equations are placed in quite new settings. The modelling through memory functionals involves constitutive equations with a joint dependence on the present values and thermal, deformation, and magnetic field histories so that, at time t, the response of the material is determined by the present values of the temperature θ, the deformation gradient F, the magnetic field H, and the temperature gradient ∇θ,
θ(t),F(t),H(t),∇θ(t)
as well as the histories, up to time *t*,
$$F^{t},H^{t},\nabla \theta^{t}.$$

Indeed, the scalar functions (internal energy, entropy, free energy) are required to be in-variant under Euclidean transformations, and this implies that the dependence on F, H, and ∇θ occurs through their invariants. As a result of the thermodynamic requirements placed by the second law inequality, explicit representations of the stress tensor, magnetization, and heat flux are established.

Next, we elaborate on differential models expressed by rate-type equations, and, hence, objective derivatives are required in the rate equations. Due to the occurrence of the time derivative, objectivity indicates that the Lagrangian description is more convenient as the starting step. A general scheme is allowed by letting an invariant stress T, the Green–Saint Venant tensor E, and a magnetic field vector be the independent variables. The generality is allowed by the joint dependence on T and E and by letting the entropy production be a constitutive function, which is quite new in the literature. Furthermore, to determine the restrictions placed by the second law, we apply a representation formula for tensors and vectors given by Equation (Equation 4). Depending on the degree of arbitrariness of the independent (magnetic) variable, the thermodynamic analysis proves that hyper-magnetoelasticity, hypo-magnetoelasticity, and dissipative magneto-viscoelasticity are fully characterized.

The relations with other approaches in the literature are given in Section 6. Here, we indicate two features of the present work. First, the modelling through memory functionals describes the instantaneous and non-instantaneous response to the deformation, magnetic field, and temperature gradient. The thermodynamic consistency is made formal in (Equation 8) and (Equation 9). Next, specific (linear) representations are established by taking the stress, magnetization, and heat flux in the forms of (Equation 10), (Equation 11), and (Equation 14). Secondly, rate equations are considered for the stress and magnetization in terms of the deformation and magnetic field. Through the representation formula, we find the results (Equation 28) for the stress T. All of the rates given by (Equation 28) are consistent with thermodynamics; for any free energy and any entropy production, we find a physically admissible rate equation. In particular, Equations (Equation 24) and (Equation 27) describe magneto-hyperelastic materials and magneto-hypoelastic materials, respectively.

**Notation.** 
*We consider a body occupying a time-dependent region $\Omega \subset E^{3}$. The motion is described by means of the function χ(X,t), providing the position vector x∈Ω=χ(R,t). The symbols $\nabla ,\nabla_{\;R}\;$ denote the gradient operator with respect to x∈Ω, X∈R. The function χ is assumed to be differentiable; hence, we can define the deformation gradient as $F=\nabla_{\;R}\;\chi$ or, in suffix notation, $F_{iK}=\partial_{X_{K}}\chi_{i}$. The invertibility of X→x=χ(X,t) is guaranteed by letting J:=detF>0. For any tensor A, we define |A| as ${\left(A\cdot A\right)}^{1/2}$.*


## 2. Balance Equations

Let v(x,t) be the velocity field on Ω×R. A superposed dot denotes the time differentiation following the motion of the body, and hence, for any function f(x,t), we have $\dot{f}=\partial_{t}f+v\cdot \nabla f$. We denote by L the velocity gradient, $L_{ij}=\partial_{x_{j}}v_{i}$, and recall that
$$\dot{F}=LF.$$

The right Cauchy–Green tensor C and the Green–Saint Venant deformation tensor E are defined by
$$C:=F^{T}F,\;E:=\frac{1}{2}\left(C-1\right),$$
where 1 is the second-order identity tensor. Moreover, D denotes the stretching tensor, D=symL, and W the spin tensor, W=skwL.

Let ε be the internal energy density (per unit mass), T the symmetric Cauchy stress, q the heat flux vector, ρ the mass density, *r* the external heat supply, and b the mechanical body force per unit mass. Let m=M/ρ be the magnetization per unit mass and H the magnetic intensity. The balance equations for mass and linear momentum are written in the form (see, e.g., [11,12,13])
$$\dot{\rho }+\rho \nabla \cdot v=0,\;\rho \dot{v}=\nabla \cdot T+\rho b+f_{M},$$
where $f_{M}$ is the force per unit volume of magnetic character,
$$f_{M}=\mu_{0}\rho \left(m\cdot \nabla \right)H-\mu_{0}\epsilon_{0}\rho \left[\dot{m}\;\times \;E+v\;\times \;\left(m\cdot \nabla \right)E\right];$$
in stationary conditions, we can take E=0. Additionally, for later convenience, we let H be the magnetic intensity at the frame locally at rest with the body. The balance of angular momentum and energy can be written in the form
$$\mathrm{skw}(T+\mu_{0}\rho H⊗m)=0,$$
$$\rho \dot{\varepsilon }=\mu_{0}\rho H\cdot \dot{m}+T\cdot L-\nabla \cdot q+\rho r$$

Let η be the entropy density and θ the absolute temperature. As for the statement of the second law of thermodynamics, we let the inequality
(1)$$\rho \dot{\eta }+\nabla \cdot \frac{q}{\theta }-\frac{\rho r}{\theta }=\sigma \geq 0$$
hold for any process compatible with the balance equations. The scalar σ, or the entropy production, is non-negative and is viewed as a constitutive function. Hence, the thermodynamic process consists of η,q,r,σ, and the other functions occurring in the balance equations.

In terms of the magnetic Gibbs free energy
$$\phi =\varepsilon -\theta \eta -\mu_{0}m\cdot H$$
the entropy inequality can be written in the form
(2)$$-\rho \left(\dot{\phi }+\eta \dot{\theta }\right)-\mu_{0}M\cdot \dot{H}+T\cdot L-\frac{1}{\theta }q\cdot \nabla \theta =\theta \sigma \geq 0.$$

Based on (Equation 2), next, we describe the magneto-viscoelasticity by the memory functionals or rate equations and examine the thermodynamic consistency.

### Representation Formula

The thermodynamic analysis usually leads to relations of the form
(3)Z·K+A·F=f,
where, to fix ideas, Z,K,A,F are second-order tensors and *f* is a scalar. If K and F are arbitrary and independent, then it follows that Z=0,A=0 (and f=0). If, instead, K and F are not independent, then we can determine the relation between Z and A through a representation formula [8,9].

Let N be a unit tensor, |N|=1. Then
$$Z=\left(Z\cdot N\right)N+Z_{⊥},\;Z_{⊥}\cdot N=0.$$

If it happens that Z·N is known, say Z·N=g, whereas $Z_{⊥}$ is unknown, then $Z_{⊥}$ can be expressed by
$$Z_{⊥}=\left(I-N⊗N\right)G=G-\left(G\cdot N\right)N,$$
where I is the fourth-order unit tensor, and G is an arbitrary second-order tensor. Once Z·N=g is given, we can write the representation formula
Z=gN+(I−N⊗N)G.

Returning to (Equation 3), we let N=K/|K| to yield
(4)$$Z=\frac{f-A\cdot F}{{\left|K\right|}^{2}}K+\left(I-\frac{K}{\left|K\right|}⊗\frac{K}{|K}\right)G.$$

A strictly analogous relation holds if the tensors are replaced with vectors.

## 3. Constitutive Assumptions

The constitutive assumptions are suggested for several purposes. First, we allow for interaction between the deformation and temperature fields with magnetization; this indicates that θ,F,H,∇θ are among the independent variables. Moreover, the time delay in response motivates the dependence on the histories $F^{t},H^{t},\nabla \theta^{t}$. Hence, we let
$$\Gamma =(\theta ,F,H,\nabla \theta ,F^{t},H^{t},\nabla \theta^{t})$$
be the set of independent variables.

The internal energy (density) ε, the entropy η, and the Gibbs free energy ϕ are invariant under a change of frame. The constitutive equations for ε,η, and ϕ are then required to provide invariant values. Now, both F and H are not invariant. Under a change of frame
$$x^{*}=c+Qx,\;QQ^{T}=1,$$F and H change as vectors,
$$F^{*}=QF,\;H^{*}=QH.$$

Instead,
$$E=\frac{1}{2}\left(F^{T}F-1\right),\;H=F^{T}H$$
are invariant in that
$$E^{*}=F^{*T}F^{*}-1=F^{T}Q^{T}QF-1=F^{T}F-1=E,\;H^{*}=F^{*T}H^{*}=F^{T}Q^{T}QH=F^{T}H=H.$$

Of course, H·H is also invariant,
$$H^{*}\cdot H^{*}=QH\cdot QH=H\cdot Q^{T}QH=H\cdot H.$$

Incidentally,
$$H\cdot H=F^{T}H\cdot F^{T}H=H\cdot \left(FF^{T}\right)H.$$

Hence, the dependence on the pair H, E does not include that on F·F, H·H. For definiteness, we let Λ be the pair of functions
$$f\left(F\right)=\frac{1}{2}F\cdot F,\;h\left(H\right)=\frac{1}{2}H\cdot H.$$

As for the dependence on ∇θ, we observe that
$$\partial_{x^{*}}\theta =Q\partial_{x}\theta \;\mathrm{or}\;\nabla^{*}\theta =Q\nabla \theta .$$

Hence, ∇θ is a vector while the referential gradient
$$\nabla_{R}\theta =\nabla \theta F,$$
is invariant.

Let θ,F,H, and ∇θ be differentiable as t∈R for any x∈Ω. Define the constant continuation of $\theta^{t_{0}},F^{t_{0}}$, and $H^{t_{0}}$ as
$${\tilde{\theta }}^{t}\left(u\right)=\{\begin{array}{cc} \theta (t_{0}), & u\in [0,t-t_{0}),\\ \theta^{t_{0}}\left(u-\left(t-t_{0}\right)\right), & u\in [t-t_{0},\infty ), \end{array}\;\tilde{\theta }\left(t\right)={\tilde{\theta }}^{t}\left(0\right),$$
and the like for F and H while ∇θ=0 on $\left[t_{0},t\right]$.

While Γ is the set of independent variables, objectivity and modelling purposes indicate that we let
$$\phi =\phi (\theta ,E,H,\nabla_{\;R}\;\theta ,\Lambda ,E^{t},H^{t},\nabla_{\;R}\;\theta^{t},\Lambda^{t}).$$

With a small abuse of notation, the dependence on time is denoted with the same symbol
ϕ(t)=ϕ(Γ(t)).

We first show that the free energy functional ϕ(Γ) is required to satisfy a minimum property. If θ,F,H, and, hence, Λ, are constant, and ∇θ=0 in the interval $\left[t_{0},t\right]$, then (Equation 2) implies that
$$\dot{\phi }\left(\tau \right)\leq 0,\;\tau \in \left(t_{0},t\right),$$
and hence,
$$\phi \left(\tilde{\theta }\left(t\right),\tilde{E}\left(t\right),\tilde{H}\left(t\right),\tilde{\Lambda }\left(t\right),0,{\tilde{E}}^{t},{\tilde{H}}^{t},\nabla_{\;R}\;{\tilde{\theta }}^{t},{\tilde{\Lambda }}^{t}\right)\leq \phi \left(\theta_{0},E_{0},H_{0},\Lambda_{0},0,E^{t_{0}},H^{t_{0}},\nabla_{\;R}\;\theta^{t_{0}},\Lambda^{t_{0}}\right),$$
where we let $\theta_{0}=\theta \left(t_{0}\right),E_{0}=E\left(t_{0}\right),H_{0}=H\left(t_{0}\right),\;\mathrm{and}\;\Lambda_{0}=\Lambda \left(t_{0}\right)$. By the continuity of the functional, as $t-t_{0}\to \infty$ we have
$$\phi \left(\theta_{0},E_{0},H_{0},0,\Lambda_{0},{\tilde{E}}^{t},{\tilde{H}}^{t},\nabla_{\;R}\;{\tilde{\theta }}^{t},{\tilde{\Lambda }}^{t}\right)\to \phi \left(\theta_{0},E_{0},H_{0},0,\Lambda_{0},E_{0}^{†},H_{0}^{†},0^{†},\Lambda_{0}^{†}\right),$$
where $E_{0}^{†},\;H_{0}^{†},\;0^{†}$, and $\Lambda_{0}^{†}$ are the constant histories with values of $E_{0},H_{0},0$, and $\Lambda_{0}$. Thus, it follows that
$$\phi \left(\theta_{0},E_{0},H_{0},0,\Lambda_{0},E_{0}^{†},H_{0}^{†},0^{†},\Lambda_{0}^{†}\right)\leq \phi \left(\theta_{0},E_{0},H_{0},0,\Lambda_{0},E_{0}^{t_{0}},H_{0}^{t_{0}},\nabla_{\;R}\;\theta^{t_{0}},\Lambda^{t_{0}}\right).$$

This is the content of the minimum property: among all histories $E^{t},H^{t},\nabla_{\;R}\;\theta^{t},$ and $\Lambda^{t}$ with the given present values of $E_{0},H_{0},0$, and $\Lambda_{0}$, none yields a smaller value of the free energy than that corresponding to the constant histories $E_{0}^{†},\;H_{0}^{†},\;0^{†}$ and $\Lambda_{0}^{†}$.

## 4. Thermodynamic Restrictions

We now compute the time derivative $\dot{\phi }$ and substitute in the entropy inequality to obtain
$$\begin{array}{c} -\rho \left(\partial_{\theta }\phi +\eta \right)\dot{\theta }-\rho \partial_{E}\phi \cdot \dot{E}-\rho \partial_{H}\phi \cdot \dot{H}-\rho \partial_{\nabla \theta }\phi \cdot \dot{\bar{\nabla \theta }}-\mu_{0}M\cdot \dot{H}-\rho \partial_{f}\phi F\cdot \dot{F}-\rho \partial_{h}\phi H\cdot \dot{H}\\ +T\cdot L-\frac{1}{\theta }q\cdot \nabla \theta -\rho d\phi \left(\Gamma \right|{\dot{E}}^{t})-\rho d\phi \left(\Gamma |{\dot{H}}^{t}\right)-\rho d\phi \left(\Gamma |{\dot{\bar{\nabla \theta }}}^{t}\right)-\rho d\phi \left(\Gamma^{t}|{\dot{f}}^{t}\right)-\rho d\phi \left(\Gamma^{t}|{\dot{h}}^{t}\right)\geq 0. \end{array}$$

Here, we have made the dependence on Λ=(f,h) and $\Lambda^{t}=\left(f^{t},h^{t}\right)$ explicit. The linearity and arbitrariness of $\dot{\bar{\nabla \theta }},\dot{\theta }$ imply
$$\partial_{\nabla \theta }\phi =0,\;\eta =-\partial_{\theta }\phi .$$

Observe that
(5)$$\dot{E}=F^{T}DF,\;\dot{H}=F^{T}L^{T}H+F^{T}\dot{H},\;F\cdot \dot{F}=\left(FF^{T}\right)\cdot L.$$

Since L=D+W, the occurrence of W in the inequality is through
$$(T+\rho F\partial_{H}\phi ⊗H-\rho \partial_{f}\phi FF^{T})\cdot W$$
and this quantity has to vanish. Now $FF^{T}\in \mathrm{Sym}$ and then $(FF^{T})\cdot W=0$ identically. Hence, we have
$$(T+\rho F\partial_{H}\phi ⊗H)\cdot W=0$$
which implies
(6)$$T+\rho F\partial_{H}\phi ⊗H\in \mathrm{Sym}.$$

We now can write the remaining inequality in the form
(7)$$\begin{array}{c} \left(T-\rho F\partial_{E}\phi F^{T}+\rho F\partial_{H}\phi ⊗H-\rho \partial_{f}\phi FF^{T}\right)\cdot D-\left(\mu_{0}M+\rho F\partial_{H}\phi -\rho \partial_{h}\phi H\right)\cdot \dot{H}\\ -\frac{1}{\theta }q\cdot \nabla \theta -\rho d\phi \left(\Gamma \right|{\dot{E}}^{t})-\rho d\phi \left(\Gamma |{\dot{H}}^{t}\right)-\rho d\phi \left(\Gamma |{\dot{\bar{\nabla \theta }}}^{t}\right)-\rho d\phi \left(\Gamma^{t}|{\dot{f}}^{t}\right)-\rho d\phi \left(\Gamma^{t}|{\dot{h}}^{t}\right)\geq 0. \end{array}$$

The linearity and arbitrariness of $D,\dot{H}$ imply
(8)$$T=\rho F\partial_{E}\phi F^{T}-\rho F\partial_{H}\phi ⊗H,\;\mu_{0}M=-\rho F\partial_{H}\phi +\rho \partial_{h}\phi \;H.$$

Inequality (Equation 7) then reduces to
(9)$$-\frac{1}{\theta }q\cdot \nabla \theta -\rho d\phi \left(\Gamma |{\dot{E}}^{t}\right)-\rho d\phi \left(\Gamma |{\dot{H}}^{t}\right)-\rho d\phi \left(\Gamma |{\dot{\bar{\nabla \theta }}}^{t}\right)\geq 0.$$

According to (Equation 6) and (Equation 8), it follows that
$$\mathrm{skw}T=\mathrm{skw}(\mu_{0}M⊗H)$$
which is just the requirement placed by the balance of angular momentum. It is worth remarking that this requirement holds merely because of the dependence of ϕ on H through $H=F^{T}H$. Instead, the dependence of ϕ on *h* leaves skwT unchanged.

Further restrictions, placed by the reduced inequality (Equation 9), follow by considering some particular cases. First, we assume the temperature is uniform at any time, $\nabla \theta^{t}=0$. Moreover, nonlinear constitutive equations for T and M are established by selecting the partial derivatives
(10)$$\partial_{E}\phi =G_{0}\left(\theta ,|E|\right)E+g\left(\theta \right)\int_{0}^{\infty }G^{\prime }\left(u\right)E\left(t-u\right)du,\;G_{0},G^{\prime }\in \mathrm{Lin},$$
(11)$$\partial_{H}\phi =-A_{0}\left(\theta ,|H|\right)H-a\left(\theta \right)\int_{0}^{\infty }A^{\prime }\left(u\right)H\left(t-u\right)du,\;A_{0},A^{\prime }\in \mathrm{Lin}.$$

Now, given G(θ,|E|), a fully symmetric fourth-order tensor-valued function, we have
$$\partial_{E}\frac{1}{2}E\cdot GE=GE+\frac{1}{2|E|}\left[E\cdot (\partial_{\left|E\right|}G)E\right]E$$
and the like for a symmetric second-order tensor-valued function A(θ,|H|). Hence, letting 1 and I be the second- and fourth-order unit tensors, we define
$$G_{0}=G+\frac{1}{2|E|}\left[E\cdot (\partial_{\left|E\right|}G)E\right]\;I,\;A_{0}=A+\frac{1}{2|H|}\left[H\cdot (\partial_{\left|H\right|}A)H\right]1,$$
so that
$$G_{0}E=\partial_{E}\;\frac{1}{2}E\cdot GE,\;A_{0}\;H=\partial_{H}\;\frac{1}{2}\;H\cdot A\;H.$$

Hence, the sought functional $\phi (\theta ,E,H,E^{t},H^{t})$ takes the form
$$\begin{array}{c} \phi =\;\frac{1}{2}E\cdot GE+g\left(\theta \right)E\cdot \int_{0}^{\infty }G^{\prime }\left(u\right)E\left(t-u\right)du\\ -\frac{1}{2}\;H\cdot A\;H-a\left(\theta \right)H\cdot \int_{0}^{\infty }A^{\prime }\left(u\right)H\left(t-u\right)du+\Phi \left(\theta ,E^{t},H^{t}\right), \end{array}$$
and the functional Φ is thus far undetermined. According to (Equation 9), the functional ϕ has to satisfy the inequality
(12)$$d\phi \left(\Gamma |{\dot{E}}^{t}\right)+d\phi \left(\Gamma |{\dot{H}}^{t}\right)\leq 0$$
and this is eventually the requirement of the unknown functional Φ.

For definiteness, we consider Φ in the form
$$\Phi \left(\theta ,E^{t},H^{t}\right)=\phi_{0}\left(\theta \right)+\frac{1}{2}g\left(\theta \right)\int_{0}^{\infty }E^{t}\left(u\right)\cdot G^{\prime }\left(u\right)E^{t}\left(u\right)\;du-\frac{1}{2}a\left(\theta \right)\int_{0}^{\infty }H^{t}\left(u\right)\cdot A^{\prime }\left(u\right)H^{t}\left(u\right)\;du$$
and, hence, the functional ϕ can be written as $\phi =\phi_{m}$,
$$\begin{array}{c} \phi_{m}:=\phi_{0}\left(\theta \right)+\frac{1}{2}E\cdot G_{\infty }E-\frac{1}{2}g\left(\theta \right)\int_{0}^{\infty }\left[E\left(t\right)-E\left(t-u\right)\right]\cdot G^{\prime }\left(u\right)\left[E\left(t\right)-E\left(t-u\right)\right]du\\ -\frac{1}{2}H\cdot A_{\infty }\;H+\frac{1}{2}g\left(\theta \right)\int_{0}^{\infty }\left[H\left(t\right)-H\left(t-u\right)\right]\cdot A^{\prime }\left(u\right)\left[H\left(t\right)-H\left(t-u\right)\right]du. \end{array}$$

It follows that
$$\begin{array}{c} d\phi \left(\Gamma |{\dot{E}}^{t}\right)=-\frac{1}{2}g\left(\theta \right)\int_{0}^{\infty }\left[E\left(t\right)-E\left(t-u\right)\right]\cdot G^{\prime }\left(u\right)\dot{E}\left(t-u\right)]du\\ =\frac{1}{2}g\left(\theta \right)\{{\left[\left(E\left(t\right)-E\left(t-u\right)\right)\cdot G^{\prime }\left(u\right)\left(E\left(t\right)-E\left(t-u\right)\right)\right]}_{0}^{\infty }\\ -\int_{0}^{\infty }\left(E\left(t\right)-E\left(t-u\right)\right)\cdot G^{\prime \prime }\left(u\right)\left(E\left(t\right)-E\left(t-u\right)\right)du\}. \end{array}$$

The assumption $G^{\prime }\left(\infty \right)=0$ implies
$$d\phi \left(\Gamma |{\dot{E}}^{t}\right)=-\frac{1}{2}g\left(\theta \right)\int_{0}^{\infty }\left[E\left(t\right)-E\left(t-u\right)\right]\cdot G^{\prime \prime }\left(u\right)\left[E\left(t\right)-E\left(t-u\right)\right]du.$$

Likewise, letting $A^{\prime }\left(\infty \right)=0$, we find
$$d\phi \left(\Gamma |{\dot{H}}^{t}\right)=\frac{1}{2}a\left(\theta \right)\int_{0}^{\infty }\left[H\left(t\right)-H\left(t-u\right)\right]\cdot A^{\prime \prime }\left(u\right)\left[H\left(t\right)-H\left(t-u\right)\right]du.$$

Hence, (Equation 12) holds if and only if
$$g\left(\theta \right)G^{\prime \prime }\left(u\right)\geq 0,\;a\left(\theta \right)A^{\prime \prime }\left(u\right)\leq 0$$
for all u∈[0,∞).

The minimum property of $\phi_{m}$ at the constant histories $E^{†},H^{†}$ holds if and only if
$$g\left(\theta \right)G^{\prime }\left(u\right)\leq 0,\;a\left(\theta \right)A^{\prime }\left(u\right)\geq 0$$
for all u∈[0,∞).

Hence, the functional $\phi_{m}$ is thermodynamically consistent if $G^{\prime }$ and $G^{\prime \prime }$, as well as $A^{\prime }$ and $A^{\prime \prime }$, have opposite types of definiteness.

### Heat Conduction

Now we let $\nabla_{\;R}\;\theta^{t}\neq 0$, and, for simplicity, we look for models where $\nabla_{\;R}\;\theta$ is independent of F and H so that the reduced dissipation inequality splits into (Equation 12) and
$$\rho \theta d\phi \left(\Gamma |\nabla_{\;R}\;{\dot{\theta }}^{t}\right)+\frac{1}{\theta }q\cdot \nabla \theta \geq 0.$$

Multiply this inequality by J=detF and observe that, using the referential heat flux $q_{R}=JF^{-T}q$, we can write
(13)$$\rho_{R}\theta d\phi \left(\Gamma |\nabla_{\;R}\;{\dot{\theta }}^{t}\right)+q_{R}\cdot \nabla_{\;R}\;\theta \leq 0.$$

For definiteness, let $q_{R}$ be given by the constitutive functional
(14)$$q_{R}\left(\theta \left(t\right),\nabla_{\;R}\;\theta^{t}\right)=\alpha \left(\theta \right)K_{0}\int_{0}^{\infty }\beta \left(u\right)\nabla_{\;R}\;\theta \left(t-u\right)du,$$
where $K_{0}$ is a positive-definite second-order tensor, while α and β are so far undetermined; we only assume β(∞)=0.

We let $\phi =\phi_{m}+\phi_{c}$ with $\phi_{c}$ taken in the form
$$\phi_{c}=\alpha \left(\theta \right)|K_{0}^{1/2}\int_{0}^{\infty }\beta \left(u\right)\nabla_{\;R}\;\theta \left(t-u\right)du{|}^{2}.$$

The minimum property of ϕ, and hence of $\phi_{c}$, at $\nabla_{\;R}\;\theta \equiv 0$ holds if and only if α>0. Now, inequality (Equation 13) results in
(15)$$\begin{array}{c} \rho_{R}\theta \left(K_{0}\int_{0}^{\infty }\beta \left(u\right)\nabla_{\;R}\;\theta \left(t-u\right)du\right)\cdot \int_{0}^{\infty }\beta \left(u\right)\nabla_{\;R}\;\dot{\theta }\left(t-u\right)du\\ +\left(K_{0}\int_{0}^{\infty }\beta \left(u\right)\nabla_{\;R}\;\theta \left(t-u\right)du\right)\cdot \nabla_{\;R}\;\theta \leq 0. \end{array}$$

An integration by parts yields
$$\begin{array}{c} \int_{0}^{\infty }\beta \left(u\right)\nabla_{\;R}\;\dot{\theta }\left(t-u\right)du=-\int_{0}^{\infty }\beta \left(u\right)\partial_{u}\nabla_{\;R}\;\theta \left(t-u\right)du=-{\left[\beta \left(u\right)\nabla_{\;R}\;\theta \left(t-u\right)\right]}_{0}^{\infty }\\ +\int_{0}^{\infty }\beta^{\prime }\left(u\right)\nabla_{\;R}\;\theta \left(t-u\right)du=\beta \left(0\right)\nabla_{\;R}\;\theta \left(t\right)+\int_{0}^{\infty }\beta^{\prime }\left(u\right)\nabla_{\;R}\;\theta \left(t-u\right)du. \end{array}$$

Hence, inequality (Equation 15) can be written in the form
(16)$$\begin{array}{c} \left(\rho_{R}\theta \beta \left(0\right)+1\right)\left(K_{0}\int_{0}^{\infty }\beta \left(u\right)\nabla_{\;R}\;\theta \left(t-u\right)du\right)\cdot \nabla_{\;R}\;\theta \left(t\right)\;\;\;\\ +\rho_{R}\theta \left(K_{0}\int_{0}^{\infty }\beta \left(u\right)\nabla_{\;R}\;\theta \left(t-u\right)du\right)\cdot \left(\int_{0}^{\infty }\beta^{\prime }\left(u\right)\nabla_{\;R}\;\theta \left(t-u\right)du\right)\leq 0. \end{array}$$

The linearity and arbitrariness of $\nabla_{\;R}\;\theta \left(t\right)$ imply that
$$\beta \left(0\right)=-\frac{1}{\rho_{R}\theta },\;\beta \left(u\right)\beta^{\prime }\left(u\right)\leq 0.$$

Since β(0)<0 and ${\left(\beta^{2}\right)}^{\prime }\leq 0$, then β(u)≤0 for all u∈[0,∞). Hence,
$$\alpha \left(\theta \right)K_{0}\beta \left(u\right)\leq 0$$
and, consistently, the kernel of the functional $q_{R}$ is negative definite.

Observe that
$$\hat{\beta }\left(u\right)=\frac{\beta (u)}{\beta (0)}\geq 0.$$

Hence, we can write $q_{R}$ in the form
$$q_{R}=-\frac{\alpha (\theta )}{\rho_{R}\theta }K_{0}\int_{0}^{\infty }\hat{\beta }\left(u\right)\nabla_{\;R}\;\theta \left(t-u\right)du.$$

At the limit of short memory, we have
$$q_{R}=-\frac{\alpha (\theta )}{\rho_{R}\theta }K_{0}\left(\int_{0}^{\infty }\hat{\beta }\left(u\right)du\right)\nabla_{\;R}\;\theta .$$

In the spatial description, the constitutive equation reads
$$q=-\frac{\alpha (\theta )}{J\rho_{R}\theta }FK_{0}\int_{0}^{\infty }\beta \left(u\right)\left(\nabla \theta F\right)\left(t-u\right)du.$$

## 5. Rate Equations in the Eulerian Description

It is a crucial point of magnetoelasticity, as well as of magneto-viscoelasticity, that the stress tensor need not be symmetric. Hence, the mechanical power T·L need not equal T·D, and, moreover,
$$T\cdot L=J^{-1}\left(FT_{RR}F^{T}\right)\cdot L=J^{-1}T_{RR}\cdot \left(F^{T}DF\right)+J^{-1}T_{RR}\cdot \left(F^{T}WF\right).$$

Since $\dot{E}=F^{T}DF$, then
$$T\cdot L=J^{-1}T_{RR}\cdot \dot{E}+J^{-1}T_{RR}\cdot \left(F^{T}WF\right).$$

We start with the Eulerian description and write the Clausius–Duhem inequality in the form
(17)$$-\rho \left(\dot{\phi }+\eta \dot{\theta }\right)+T\cdot L-\mu_{0}M\cdot \dot{H}-\frac{1}{\theta }q\cdot \nabla \theta =\theta \sigma \geq 0.$$

Since we look for rate equations, objectivity indicates that the independent variables are invariant so that their time derivatives are invariant too. Hence, we assume that
ϕ=ϕ(θ,T,E,H),
where T is a stress-like variable to be identified. Consequently, the Clausius–Duhem inequality (Equation 17) can be written as
$$-\rho \left(\partial_{\theta }\phi +\eta \right)\dot{\theta }-\rho \partial_{T}\phi \cdot \dot{T}-\rho \partial_{E}\phi \cdot \dot{E}-\rho \partial_{H}\phi \cdot \dot{H}+T\cdot L-\mu_{0}M\cdot \dot{H}\geq 0.$$

Since $\dot{H}=F^{T}L^{T}H+F^{T}\dot{H}$, then we have
$$\begin{array}{c} -\rho \left(\partial_{\theta }\phi +\eta \right)\dot{\theta }-\rho \partial_{T}\phi \cdot \dot{T}-\rho \partial_{E}\phi \cdot \dot{E}+\left(T-\rho F\partial_{E}\phi F^{T}-\rho H⊗F\partial_{H}\phi \right)\cdot \left(F^{-T}\dot{E}F^{-1}\right)\\ +\left(T-\rho H⊗F\partial_{H}\phi \right)\cdot W-\left(\rho F\partial_{H}\phi +\mu_{0}M\right)\cdot \dot{H}\geq 0. \end{array}$$

Hence, it follows
(18)$$\eta =-\partial_{\theta }\phi ,$$
(19)$$\mu_{0}M=-\rho F\partial_{H}\phi ,$$
(20)$$\mathrm{skw}(T+\mu_{0}H⊗M)=0.$$

Thus, we replace $-\rho F\partial_{H}\phi$ with $\mu_{0}M$ and write the remaining inequality in the form
$$-\rho \partial_{T}\phi \cdot \dot{T}+\left(-\rho \partial_{E}\phi +J^{-1}T_{RR}+\mu_{0}F^{-1}H⊗F^{-1}M\right)\cdot \dot{E}\geq 0$$

This result indicates that we let
(21)$$T:=T_{RR}+J\mu_{0}F^{-1}H⊗F^{-1}M$$
so that we have
(22)$$-\rho_{R}\partial_{T}\phi \cdot \dot{T}+\left(T-\rho_{R}\partial_{E}\phi \right)\cdot \dot{E}=J\theta \sigma .$$

Observe that $F^{-1}H$ and $F^{-1}M$ are invariant vectors. To yield this
$${\left(F^{-1}H\right)}^{*}={\left(QF\right)}^{-1}QH=F^{-1}Q^{-1}QH=F^{-1}H$$
and the like for $F^{-1}M$. Hence, T is an invariant tensor. Moreover, by letting $M=JF^{-1}M$ we can write (Equation 19) in the form
(23)$$\mu_{0}\;M=-\partial_{H}\phi .$$

If $\dot{T}$ and $\dot{E}$ are independent, then we obtain
(24)$$\partial_{T}\phi =0,\;T=\rho_{R}\partial_{E}\phi$$
and then σ=0. The equations in (Equation 24) are said to characterize magneto-hyperelastic materials. The results (Equation 23) and (Equation 24) allow us to write the incremental relations
(25)$$\mu_{0}\dot{M}=-\rho_{R}\partial_{E}\partial_{H}\phi \;\dot{E}-\rho_{R}\partial_{H}\partial_{H}\phi \;\dot{H},$$
(26)$$\dot{T}=\rho_{R}\partial_{E}\partial_{E}\phi \;\dot{E}+\rho_{R}\partial_{H}\partial_{E}\phi \;\dot{H}.$$

If, again, σ=0, but
$$\partial_{T}\phi \neq 0,\;\partial_{E}\phi \neq 0,$$
then we have
(27)$$-\rho_{R}\partial_{T}\phi \cdot \dot{T}+\left(T-\rho_{R}\partial_{E}\phi \right)\cdot \dot{E}=0;$$

Equation (Equation 27) is said to characterize magneto-hypoelastic materials. In general, we can express $\dot{T}$ via the representation formula,
$$\dot{T}=\left(\dot{T}\cdot N\right)N+\left(I-N⊗N\right)G,$$
where G is any second-order tensor. Let $N=\partial_{T}\phi /\left|\partial_{T}\phi \right|$. According to (Equation 27), we have
$$\dot{T}=\frac{\left(T-\rho_{R}\partial_{E}\phi \right)\cdot \dot{E}}{\rho_{R}{\left|\partial_{T}\phi \right|}^{2}}\partial_{T}\phi +\left(I-\frac{\partial_{T}\phi }{|\partial_{T}\phi |}⊗\frac{\partial_{T}\phi }{|\partial_{T}\phi |}\right)G$$
or
$$\dot{T}=\frac{1}{\rho_{R}{\left|\partial_{T}\phi \right|}^{2}}\left[\partial_{T}\phi ⊗\left(T-\rho_{R}\partial_{E}\phi \right)\right]\dot{E}+\left(I-\frac{\partial_{T}\phi }{|\partial_{T}\phi |}⊗\frac{\partial_{T}\phi }{|\partial_{T}\phi |}\right)G.$$

If σ≠0, then we have
$$-\rho_{R}\partial_{T}\phi \cdot \dot{T}+\left(T-\rho_{R}\partial_{E}\phi \right),\cdot \dot{E}=J\theta \sigma$$
and the representation formula for $\dot{T}$ generalizes to
(28)$$\dot{T}=\frac{\left(T-\rho_{R}\partial_{E}\phi \right)\cdot \dot{E}-J\theta \sigma }{\rho_{R}{\left|\partial_{T}\phi \right|}^{2}}\partial_{T}\phi +\left(I-\frac{\partial_{T}\phi }{|\partial_{T}\phi |}⊗\frac{\partial_{T}\phi }{|\partial_{T}\phi |}\right)G.$$

Definite forms of (Equation 28) are now established by having in mind fluid or solid behaviours.

### 5.1. Fluids

The interaction between deformation and magnetization can be modelled by letting both ϕ and σ depend on T and H. For definiteness, we let
$$\rho_{R}\phi =\rho_{R}\phi_{0}\left(\theta \right)+\frac{1}{2}{\alpha |T|}^{2}-\frac{1}{2}\xi {\left|H\right|}^{2},\;\alpha ,\xi >0,$$
and
$$J\theta \sigma ={\beta |T|}^{2}+\gamma {\left|H\right|}^{2},\;\beta ,\gamma >0.$$

Equations (Equation 18) and (Equation 19) hold. According to (Equation 19), the dependence of ϕ on H results in the magnetization relation,
$$\mu_{0}\;M:=\mu_{0}JF^{-1}M=\xi \;H.$$

Since
$$\partial_{E}\phi =0,\;\rho_{R}\partial_{T}\phi =\alpha T$$
then (Equation 28) can be written in the form
$$\dot{T}=\frac{T⊗T}{{\alpha |T|}^{2}}\dot{E}-\frac{\beta }{\alpha }T-\frac{{\gamma |H|}^{2}}{{\alpha |T|}^{2}}T+\left(I-\frac{T}{\left|T\right|}⊗\frac{T}{\left|T\right|}\right)G.$$

If, for definiteness, we let $G=\dot{E}/\alpha$, then it follows
(29)$$\dot{T}=\frac{1}{\alpha }\dot{E}-\frac{\beta }{\alpha }T-\frac{{\gamma |H|}^{2}}{{\alpha |T|}^{2}}T.$$

In particular, the choice $\gamma =\gamma_{0}{\left|T\right|}^{2}$ yields
(30)$$\dot{T}=a\dot{E}-bT,\;a=1/\alpha >0,\;b=\beta /\alpha +\gamma_{0}{\left|H\right|}^{2}/\alpha >0,$$
that is a generalization of the Maxwell (fluid) model. Given $T_{0}=T\left(0\right)$, if |H| is considered known, then the linear ODE (Equation 30) with non-constant coefficients can be solved on (0,T) to obtain
(31)$$T\left(t\right)=T_{0}\exp\left[-\overset{t}{\underset{0}{\int }}b\left(\lambda \right)d\lambda \right]+\int_{0}^{t}a\left(\tau \right)\exp\left[-\overset{t}{\underset{\tau }{\int }}b\left(\lambda \right)d\lambda \right]\dot{E}\left(\tau \right)d\tau .$$

If $\tilde{E}$ is a given past history on (−∞,0], we assume
$$T_{0}=\int_{-\infty }^{0}a\left(\tau \right)\exp\left[-\overset{0}{\underset{\tau }{\int }}b\left(\lambda \right)d\lambda \right]\dot{\tilde{E}}\left(\tau \right)d\tau .$$

By substituting $T_{0}$ in (Equation 31), we have
$$T\left(t\right)=\int_{-\infty }^{t}a\left(\tau \right)\exp\left[-\overset{t}{\underset{\tau }{\int }}b\left(\lambda \right)d\lambda \right]\dot{E}\left(\tau \right)d\tau =\int_{0}^{\infty }a\left(t-s\right)\exp\left[-\overset{s}{\underset{0}{\int }}b\left(\xi \right)d\xi \right]\dot{E}\left(t-s\right)ds,$$
where $E\left(\tau \right)=\tilde{E}\left(\tau \right),\tau \leq 0$.

We observe that the memory kernel of (Equation 31) has the form
$$G\left(\tau ,t-\tau \right):=a\left(\tau \right)\exp\left[-\overset{t}{\underset{\tau }{\int }}b\left(\lambda \right)d\lambda \right]=a\left(\tau \right)\exp\left[-\overset{t-\tau }{\underset{0}{\int }}b\left(\xi \right)d\xi \right]$$
which describes an aging effect as the *G* function changes over time due to the presence of the a(τ) factor. We obtain a standard memory kernel
$$G\left(t-\tau \right)=a_{0}\exp\left[-\overset{t}{\underset{\tau }{\int }}b\left(\lambda \right)d\lambda \right]$$
if $a=a_{0}$ is constant.

Things are different if σ is viewed as a viscous term in the form
$$J\theta \sigma =\nu |\dot{E}{|}^{2}+\zeta {\left|\dot{H}\right|}^{2},\;\nu ,\zeta >0.$$

It follows that
(32)$$\dot{T}=\frac{T⊗T}{{\alpha |T|}^{2}}\dot{E}-\frac{\nu |\dot{E}{|}^{2}+\zeta {\left|\dot{H}\right|}^{2}}{{\alpha |T|}^{2}}T+\left(I-\frac{T}{\left|T\right|}⊗\frac{T}{\left|T\right|}\right)G.$$

If $G=\dot{E}/\alpha$, we have
$$\dot{T}=\frac{\dot{E}}{\alpha }-\frac{\nu |\dot{E}{|}^{2}+\zeta {\left|\dot{H}\right|}^{2}}{{\alpha |T|}^{2}}T.$$

Observe that
$$\dot{H}={\dot{F}}^{T}H+F^{T}\dot{H}=F^{T}\left(L^{T}H+\dot{H}\right)$$

Now,
$$L^{T}H+\dot{H}=\overset{\circ }{\;H}+DH,$$
where $\overset{\circ }{\;H}$ is the corotational derivative, $\overset{\circ }{\;H}=\dot{H}-WH$. Hence, we have
$$|\dot{H}{|}^{2}=\left(\overset{\circ }{\;H}+DH\right)\cdot FF^{T}\left(\overset{\circ }{\;H}+DH\right).$$

Equation (Equation 32) can then be written in the form
$$\dot{T}=\frac{T⊗T}{{\alpha |T|}^{2}}\dot{E}-\frac{\nu |\dot{E}{|}^{2}}{{\alpha |T|}^{2}}T-\frac{\zeta \left(\overset{\circ }{\;H}+DH\right)\cdot FF^{T}\left(\overset{\circ }{\;H}+DH\right)}{{\alpha |T|}^{2}}T+\left(I-\frac{T}{\left|T\right|}⊗\frac{T}{\left|T\right|}\right)G.$$

### 5.2. Solids

Solids are characterized by a stress dependence such that, asymptotically, $T=G_{\infty }E$. Hence, we formally replace the T of the fluid model with $T-G_{\infty }E$. Define
$$\rho_{R}\phi =\rho_{R}\phi_{0}\left(\theta \right)+\frac{1}{2}EG_{\infty }E+\frac{1}{2}\left(T-G_{\infty }E\right)\cdot A\left(T-G_{\infty }E\right)-\frac{1}{2}H\cdot \Xi H$$
and
$$J\theta \sigma =\left(T-G_{\infty }E\right)\cdot \left[{\left(\beta +H\cdot \Gamma H\right)}^{-1}A\left(T-G_{\infty }E\right)\right],\;\beta >0,$$
where $\Xi ,\Gamma \in {\mathrm{Sym}}^{+}$ and $A,G_{\infty }$ are positive-definite fourth-order tensors. Observe that
$$\rho_{R}\partial_{H}\phi =-\Xi \;H,\;\rho_{R}\partial_{T}\phi =A\left(T-G_{\infty }E\right),\;\rho_{R}\partial_{E}\phi G_{\infty }E-G_{\infty }^{T}A\left(T-G_{\infty }E\right).$$

Moreover, according to (Equation 19), it follows that $M:=JF^{-1}M=\mu_{0}^{-1}\Xi \;H$.

We now apply the representation (Equation 28) by letting
$$N=\frac{\partial_{T}\phi }{|\partial_{T}\phi |}=\frac{A(T-G_{\infty }E)}{\left|A(T-G_{\infty }E)\right|}.$$

Within (I−N⊗N)G, the representation formula yields
$$\begin{array}{c} \dot{T}≃\frac{\left[T-G_{\infty }E+G_{\infty }^{T}A\left(T-G_{\infty }E\right)\right]\dot{E}-\left(T-G_{\infty }E\right)\cdot {\left[\beta +H\cdot \Gamma H\right]}^{-1}A\left(T-G_{\infty }E\right)}{\left|A(T-G_{\infty }E)\right|}N\\ ≃\frac{A\left(T-G_{\infty }E\right)\cdot \left[A^{-1}+G_{\infty }E\right)]\dot{E}-A\left(T-G_{\infty }E\right)\cdot {\left[\beta +H\cdot \Gamma H\right]}^{-1}\left(T-G_{\infty }E\right)}{\left|A(T-G_{\infty }E)\right|}N. \end{array}$$

Hence, we have
$$\dot{T}=\left(N⊗N\right)\left[A^{-1}+G_{\infty }\right]\dot{E}-\left(N⊗N\right){\left[\beta +H\cdot \Gamma H\right]}^{-1}\left(T-G_{\infty }E\right)+\left(I-N⊗N\right)G.$$

Choosing, e.g.,
$$G=\left[A^{-1}+G_{\infty }\right]\dot{E}-{\left[\beta +H\cdot \Gamma H\right]}^{-1}\left(T-G_{\infty }E\right)$$
we find
$$\dot{T}=\left[A^{-1}+G_{\infty }\right]\dot{E}-{\left[\beta +H\cdot \Gamma H\right]}^{-1}\left(T-G_{\infty }E\right),$$
whence
(33)$$\dot{T}-G_{\infty }\dot{E}+{\left[\beta +H\cdot \Gamma H\right]}^{-1}\left(T-G_{\infty }E\right)=A^{-1}\dot{E}.$$

Equation (Equation 33) shows that $T-G_{\infty }E$ evolves with a relaxation time
τ=β+H·ΓH.

Moreover, if $\dot{E}=0$, then, asymptotically, we have
$$T=G_{\infty }E,$$
as we expected for a solid model. We finally note that (Equation 33) takes the usual form
(34)$$\dot{T}=G_{0}\dot{E}-\frac{1}{\tau }\left(T-G_{\infty }E\right)$$
after letting $G_{0}=A^{-1}+G_{\infty }$.

### 5.3. A One-Dimensional Example

Restrict attention to one-dimensional models associated with strain, applied traction, and magnetic field in the direction e such that
E=Ee⊗e,T=Te⊗e,H=He.

The symbol *T* for the component of T is consistent with the *engineering stress* considered in the literature to be the ratio of the axial force over the reference area. Moreover, let $G_{\infty }=G_{\infty }I$, $G_{0}=G_{0}I$, and γ=e·Γe; Equation (Equation 34) can be written as
$$\dot{T}=G_{0}\dot{E}-\frac{1}{\tau }\left(T-G_{\infty }E\right),\;\tau =\beta +\gamma H^{2},$$
where $G_{0},G_{\infty },\beta$, and γ>0. Assume $G_{\infty }=G_{0}\sqrt{\tau }/4$, β=0.1, and γ=1. Then, the traction response *T* for a given sinusoidal strain $E\left(t\right)=\lambda \sin\left(\frac{\pi }{20}t\right)$ is plotted in Figure 1 under different values of the magnetic fields *H*. These results are in agreement with [6] Figures 4 and 5 as they predict that the increase in the magnetic field changes the orientation of the loops and widens the hysteresis due to greater energy dissipation.

## 6. Relation to Other Approaches

Magneto-viscoelasticity is a broad subject that accounts for the interaction between a magnetic field and deformation while both elastic and dissipative effects are allowed. In this framework, various approaches have been developed. The great majority of them can be characterized according to the modelling of dissipation.

In [11,14,15,16,17], mechanical viscous effects are described by assuming the existence of an intermediate configuration that is related to the current configuration by an elastic deformation and to the initial configuration by a purely viscous deformation. Hence, the deformation gradient is given a multiplicative decomposition
$$F=F_{e}F_{v}.$$

Instead, the magnetic induction is assumed in the form
$$B=B_{e}+B_{v}.$$

The Cauchy–Green tensor C is considered in the form
$$C=F^{T}F=F_{v}^{T}F_{e}^{T}F_{e}F_{v}=F_{v}^{T}C_{e}F_{v};$$
hence, both $C_{e}$ and $C_{v}$ are defined, but $C\neq C_{e}C_{v}$. As is often the case in non-equilibrium processes, the distinction between $B_{e}$ and $B_{v}$ is based on the observation that, upon the sudden application of a constant magnetic induction, the magnetic field generated inside the material starts from an initial non-equilibrium value and then evolves to approach an equilibrium value. To model these effects, the existence of a dissipation mechanism is assumed for the magnetic induction as well. The additive decomposition of B is further motivated by the vector character B.

The entropy inequality in [11], Equation (Equation 15),
$$-\rho \dot{\Psi }+T\cdot D-M\cdot \dot{B}\geq 0$$
is consistent with (Equation 2) in that
$$M\cdot \dot{B}=M\cdot \mu_{0}\left(\dot{M}+\dot{H}\right)$$
and, hence, we have
$$-\rho \dot{\psi }+T\cdot D-\mu_{0}M\cdot \dot{H}\geq 0,\;\rho \psi =\rho \Psi +\frac{1}{2}\mu_{0}M^{2}.$$

Next, a free energy Ω is considered to be a function of the invariant fields $C,C_{v},B,B_{v}$; we observe that FB of [11] is just the magnetic induction here denoted by B. A reduced dissipation inequality then follows in the form
(35)$$\partial_{C_{v}}\Omega \cdot {\dot{C}}_{v}+\partial_{B_{v}}\Omega \cdot {\dot{B}}_{v}\leq 0.$$

Though the approach is deeply different from ours, bearing in mind equation (Equation 12), it seems natural to view the roles of $C_{v}$ and $B_{v}$ as the analogues of the dependence on the histories of E and H.

### 6.1. Incremental Magnetoelastic Equations

A simpler model is established in [18] in terms of the Lagrangian counterparts of B and H, i.e., $B=JF^{-1}B$ and $H=F^{T}H$, denoted by $B_{l}$ and $H_{l}$ in [18]. Differently from the present approach, Otténio et al. start with a “modified free energy function” Ω(F,B) such that
(36)$$T_{R}=\partial_{F}\Omega ,\;H=\partial_{B}\Omega ;$$
in components
$$T_{iK}^{R}=\partial_{F_{iK}}\Omega ,\;H_{P}=\partial_{B_{P}}\Omega .$$

The magnetic field and the deformation are then supposed to undergo changes. In the linear approximation, the changes ΔF,ΔB, and $\Delta T_{R},\Delta H$ are related in the form
$$\Delta T_{R}=A\;\Delta F+\Gamma \;\Delta B,\;\Delta \;H=\Gamma \;\Delta F+K\;\Delta B,$$
where
$$A=\partial_{F}\partial_{F}\Omega ,\;\Gamma =\partial_{F}\partial_{B}\Omega ,\;K=\partial_{B}\partial_{B}\Omega .$$
in components,
$$A_{iKjP}=\partial_{F_{iK}}\partial_{F_{jP}}\Omega ,\;\Gamma_{iKP}=\partial_{F_{iK}}\partial_{B_{P}}\Omega =\Gamma_{PiK},\;K_{PQ}=\partial_{B_{P}}\partial_{B_{Q}}\Omega .$$

Now, we look at the increments as occurring smoothly in time ($C^{1}$ functions), in the linear approximation
$$\Delta T_{R}={\dot{T}}_{R}\Delta t,$$
and the like for the other terms, so that we can write
(37)$${\dot{T}}_{R}=A\;\dot{F}+\Gamma \;\dot{B},\;\dot{H}=\Gamma \;\dot{F}+K\;\dot{B}.$$

Equations (Equation 37) are formally equal to the relations in (23) of [18] where ${\dot{T}}_{R},\dot{F},\dot{B}$, and $\dot{H}$ are said to be infinitesimal increments. There is no abuse in regarding ${\dot{T}}_{R},\dot{F},\dot{B}$, and $\dot{H}$ as time derivatives. With this view the formulas in (Equation 37) have some similarity with Equations (Equation 25) and (Equation 26). Indeed, the similarity is easily clarified once we observe that the starting assumption (12) of [18] can be derived here from the entropy inequality as the elastic part of the constitutive equations.

### 6.2. Visco-Hyperelastic Constitutive Modelling

Mainly in connection with fluid-structure coupling problems, the stress tensor is often expressed in terms of both the right and the left Cauchy–Green tensors $F^{T}F,FF^{T}$ (see, e.g., [19,20] and the references therein). The corresponding approaches lead to neither memory functionals nor rate equations, yet application of the idea underlying the visco-hyperelastic models within a simple version of the present setting is of interest.

For simplicity, we neglect the heat conduction (q=0) and the dependence on the temperature gradient (∇θ). Let θ,F,H, and D be the independent variables. Since the free energy ϕ is invariant, we assume the particular dependence ϕ=ϕ(θ,E,H,D). We then prove the thermodynamic consistency of a stress tensor in the form
(38)$$T=\hat{T}\left(\theta ,F,H\right)+T^{d}\left(D\right)$$
where
$$T^{d}\in \mathrm{Sym},\;T^{d}\left(D\right)\to 0\;\mathrm{as}\;D\to 0.$$

Consider the entropy inequality (Equation 2). Upon computation of $\dot{\phi }$, we have
$$-\rho \left(\partial_{\theta }\phi +\eta \right)\dot{\theta }-\rho \partial_{E}\phi \cdot \dot{E}-\rho \partial_{H}\phi \cdot \dot{H}-\rho \partial_{D}\phi \cdot \dot{D}+\hat{T}\cdot L+T^{d}\cdot D-\mu_{0}M\cdot \dot{H}=\theta \sigma \geq 0.$$

The arbitrariness (and linearity) of $\dot{\theta }$ and $\dot{D}$ implies
$$\eta =-\partial_{\theta }\phi ,\;\partial_{D}\phi =0.$$

In view of (Equation 5), since L=D+W, we can write the inequality in the form
(39)$$\begin{array}{c} \left(\hat{T}-\rho F\partial_{E}\phi F^{T}-\rho H⊗F\partial_{H}\phi \right)\cdot D+T^{d}\cdot D+\left(\hat{T}+\rho F\partial_{H}\phi ⊗H\right)\cdot W\;\\ -\left(\mu_{0}M+\rho F\partial_{H}\phi \right)\cdot \dot{H}=\theta \sigma \geq 0. \end{array}$$

The arbitrariness of W∈Skw and $\dot{H}$ implies
(40)$$\hat{T}+\rho F\partial_{H}\phi ⊗H\in \mathrm{Sym}$$
(41)$$\mu_{0}M=-\rho F\partial_{H}\phi ,$$
where the relation in (Equation 41) is formally equal to (Equation 6). Similar to the arbitrariness of D in (Equation 39), we conclude that the linear part is required to vanish, whence
(42)$$\mathrm{sym}[\hat{T}-\rho F\partial_{E}\phi F^{T}-\rho H⊗F\partial_{H}\phi ]=0,$$
while the nonlinear part is non-negative, $T^{d}\cdot D\geq 0$. According to (Equation 40)–(Equation 42), we can write the stress $\hat{T}$ in the form
(43)$$\hat{T}=\rho F\partial_{E}\phi F^{T}-\mu_{0}H⊗M,\;\hat{T}+\mu_{0}H⊗M\in \mathrm{Sym}.$$

If
$$T^{d}=2\mu D+\lambda \left(\mathrm{tr}\;D\right)1$$
then
$$0\leq \theta \sigma =T^{d}\cdot D=2\mu D\cdot D+\lambda {\left(\mathrm{tr}\;D\right)}^{2}$$
whence we obtain the classical inequalities μ≥0,2μ+3λ≥0 of the viscosity coefficients.

Note that the tensor $\hat{T}+\mu_{0}H⊗M$ of (Equation 43) is the Eulerian analogue of the tensor T introduced in Section 4.

### 6.3. Rheological Equations and Relaxing Media

The use of E and H in the present models is motivated by the invariance character so that $\dot{E}$ and $\dot{H}$ are also invariant and then objective. The Eulerian descriptions might involve the left Cauchy–Green tensor $FF^{T}$, rather than the right one $F^{T}F=1+2E$. In this connection, we observe that, if ϕ depends on E through scalar invariants, then $\partial_{E}\phi$ in (Equation 43) is replaced by one or more scalars, say *g*, and
$$F\partial_{E}\phi F^{T}=gFF^{T},$$
thus providing the dependence of $\hat{T}$ on the left Cauchy–Green tensor $FF^{T}$.

The elastic properties of nonlinear materials are often modelled in terms of $FF^{T}$, while dissipativity is mainly described through the stretching tensor D. This is the case, e.g., in [21] where the Cauchy stress T is given the form
(44)$$T=T_{E}\left(FF^{T}\right)+T_{V},$$
where $T_{V}$ is governed by the rate equation
$$T_{V}+\lambda DT_{V}=\nu D,$$
in which D denotes the objective derivative; ref. [21] considers the Oldroyd derivative, ${\overset{▽}{\;T}}_{V}={\dot{T}}_{V}-LT_{V}-T_{V}L^{T}$, the Jaumann derivative ${\overset{\circ }{\;T}}_{V}={\dot{T}}_{V}-WT_{V}+T_{V}W$, and the Cotter–Rivlin derivative ${\overset{△}{\;T}}_{V}={\dot{T}}_{V}+L^{T}T_{V}+T_{V}L$. Equation (Equation 44) is the analogue of (Equation 38).

It is worth remarking that in Oldroyd’s model of rheological equations for fluids, the viscous stress $T_{V}$ is subject to Equation ([22], Equation (59))
$$T_{V}+\lambda {\overset{▽}{\;T}}_{V}=2\mu D+2\mu \tau \overset{▽}{\;D}.$$

Viscoelastic fluid theories have also been established in terms of the left *relative* Cauchy–Green tensor, defined by
$$B_{t}\left(\tau \right)=F_{t}\left(\tau \right)F_{t}{\left(\tau \right)}^{T},\;F_{t}\left(\tau \right)=F\left(\tau \right)F^{-1}\left(t\right).$$

The constitutive equation is taken in the form [23]
$$T=\eta B_{1}+\beta B_{1}^{2}+\nu B_{2},$$
where
$$B_{1}\left(t\right)=\frac{d}{d\tau }B_{t}\left(\tau \right){{|}_{\tau =t},\;B_{2}\left(t\right)=\frac{d^{2}}{d\tau^{2}}B_{t}\left(\tau \right)|}_{\tau =t}.$$

## 7. Conclusions

The interaction between the deformation and magnetic field leads to a broad spectrum of phenomena. The most common scheme is merely that of stress and magnetization induced by the deformation gradient and the magnetic field, as is described by the linearized Equations (Equation 37) [18] or (Equation 25) and (Equation 26). However, the broad spectrum motivates an investigation of magneto-viscoelasticity, where both the equilibrium and dissipative properties of the interaction are modelled. This paper develops an extensive thermodynamically consistent framework and establishes new models by letting the constitutive properties involve memory functionals or rate equations.

The modelling through memory functionals involve both a dependence on the present value and the history of deformation, magnetic field, and temperature gradient. We then find that the relations for stress and magnetization are given by derivatives of the free energy with respect to the present values of the strain and magnetic field. Instead, the history dependence allows the description of the dissipative effects of the stress, magnetization, and heat conduction. Some relevant examples of the functionals are determined; Equation (Equation 11) is a linearized functional for the magnetization in terms of the history of the magnetic field.

Materials with a non-instantaneous response are often described through fractional-order derivatives [24,25,26] in place of the memory functional. This involves an unbounded kernel of the power-law form, which makes the thermodynamic consistency questionable. However, we are unaware of genuine models with memory functionals for the magnetic field through fractional-order derivatives.

The approach through rate equations turn out to be advantageous thanks as well to some ideas and results established in [8], in which the entropy production is viewed as a constitutive function, while a representation formula yields the general result (Equation 28) for solids and the analogue for fluids. The completeness of the scheme allows us to find the stress rate in magneto-hypoelasticity, magneto-hyperelasticity, and (dissipative) magneto-viscoelasticity. Equation (Equation 28) yields the rate $\dot{T}$ of the stress T in terms of the rate $\dot{E}$; the rate is characterized by the free energy ϕ(θ,T,E,H) and the entropy dissipation σ. The example devised in Section 5.3 predicts that the increase in the magnetic field broadens the hysteretic loops (see Figure 1). This is consistent with the view that the dissipation increases proportionally to the friction caused by inter-particle magnetic attraction [27]. The dual constitutive assumption ϕ=ϕ(θ,E,M,H) models the hysteresis effects in ferromagnetic materials, parameterized by the temperature θ and the strain E, as well as with hysteresis in ferroelectrics [28].

By comparing the two approaches, we can say that the dependence on histories proves convenient for linear models (non-linearities involve multiple integrals and, hence, are technically quite difficult to manage). Rate-type models are more flexible and easily allow non-linear models. However, both approaches have proved to be thermodynamically consistent.

The analysis in Section 6 of some of the approaches developed in the literature allows us to find analogies to, and differences from, the present scheme. First, there are approaches in which the strain is split into elastic and viscous parts; the present description of deformation is decomposition-free. Secondly, as expected, the instantaneous response of memory functionals or the non-dissipative version of the rate-type equations yields the analogue of magnetoelastic models (incremental equations). Thirdly, with the restriction to the instantaneous response of the magnetization, it is shown that a model holds true in which the additive terms of the stress describe the dissipative effects of the deformation within an Eulerian framework.

## Figures and Tables

**Figure 1 materials-15-06699-f001:**
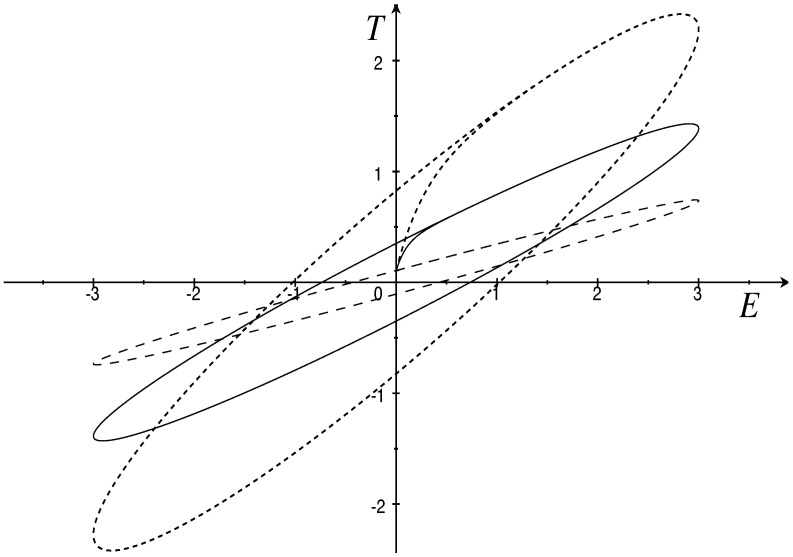
Cyclic responses in the (E,T)-plane under sinusoidal stretching with λ=3, $G_{0}=4$ and H=0.1 (dashed), H=0.4 (solid), H=0.7 (dotted).

## Data Availability

Not applicable.

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
