# Peer review of "Magneto-Viscoelastic Materials: Memory Functionals and Rate Equations"

_materials, 2022, doi:10.3390/ma15196699_

Round 1

Reviewer 1 Report

This paper investigates the magnetite-viscoelastic materials in which interaction between deformation and magnetic field leads to a wide spectrum of phenomena. Authors, developed an extensive thermodynamically-consistent framework and establishes new models by letting the constitutive properties involve memory functionals or rate equations.

The paper is very well-written. There are minor typos in the text (e.g., line 132 --> 
yields the "geenral" result for the stress rate in ). Authors must carefully proof-read the manuscript. 

On major comment in this paper is the lack of visuals. I would strongly suggest to generate plots from the developed constitutive models (consider interesting conditions) to better explain the competing nature of these materials while experiencing 
deformation and magnetic field.  Authors must address this for this paper to be publishable. Plots can also show the reduced version of the developed constitutive models (elasticity -->0).     

Reviewer 2 Report

The authors modeled the reversible changes and the dissipative effects for magneto-viscoelastic materials. The overall idea is meaningful. However, the manuscript needs substantial revision before final submission to the journal.

(1) The abstract is not clear. For example, the reason why two approaches are used is missed. Furthermore, the meaning of the sentence ‘As shown by the thermodynamic analysis, magneto-hyperelastic materials, magneto-hypoelastic materials, and various forms of magneto-viscoelastic behaviour emerge’ is not clear. There’re no conclusions. It is necessary to rewrite the abstract.

(2) In the introduction, the background and significance of this study are not introduced. It does not provide a comprehensive overview of the topic, and there are insufficient references. The difference between the present study and the previous references is not exhibited. The introduction ought to be expanded.

(3) There’s a lack of overall description about the modeling procedure. Relationships between different sections are not clear and thus, it is difficult for readers to read.

(4) Many equations do not have numbers. What does (12)2 represent?

(5) The main innovation of this study is not exhibited in detail.

(6) The authors presented the model. However, results about the model are not given. Merits and demerits of the two approaches are not clear.

(7) Revisions to the document in terms of language, grammar, structure and content are needed.

Reviewer 3 Report

Please, check the spelling, e.g.

Line 132 - "geenral" - "general"?

Reviewer 4 Report

A theory is developed to describe Reversible and dissipative effects in magneto-viscoelastic materials as modeled using memory functionals and differential rate equations. The overall development of the theory appears to be sound, but the particulars of the special cases considered are circumspect and possibly not the best examples to discuss. For instance, the right Cauchy-Green tensor is used as a specific case. This tensor is not materially objective in that it does not appropriately maintain fluid properties under an orthogonal transformation of coordinate systems. That is why it is seldom used in fluid mechanics applications where common instrumentation requires analysis via curvilinear coordinate systems. The left Cauchy-Green tensor is therefore used in these applications because it is materially objective: C* = QF(QF)^T = QFF^TQ^T = QCQ^T. In my opinion, the manuscript would be much more interesting if the particular cases examined were expanded and discussed more in depth. As it is, I feel the impact of the manuscript is rather limited, which was definitely not the intention of the authors.  There are other models and theories in the literature for this class of materials, and some are more compelling than the present case, in my opinion. I do not get a strong feeling that the present discussion offered a game-changing breakthrough in the subject, but perhaps more examples would convince me otherwise.

Round 2

Reviewer 2 Report

(1) Changes to the manuscript cannot be easily viewed by the reviewers. Any revisions to the manuscript should be marked up using the “Track Changes” function.

(2) Responses to the reviewer’s questions are not clear and comprehensive. For example, the reviewer suggested the authors to add contents about the background and significance of this study in the introduction. However, the authors did not add in the manuscript or provide explanations in the response. Logical relationships between different sections are not provided. Revisions to the document in terms of language, grammar, structure and content are still needed.

        (3) The authors stated that they used models with non-instantaneous response. However, reversible changes were described by instantaneous response in the model based on memory functionals. This is confusing.

Reviewer 4 Report

The appropriate time derivative of the LCG tensor is given as Oldroyd's B convected derivative in the original article of Oldroyd below. It is objective. It has been extensively used in continuum fluid mechanics.

https://doi.org/10.1098/rspa.1950.0035
